# Proteome partitioning constraints in long-term laboratory evolution

Matteo Mori [1], Vadim Patsalo [2], Christian Euler[3], James R. Williamson [2] & Matthew Scott [4] ✉

Adaptive laboratory evolution experiments provide a controlled context in which the dynamics of selection and adaptation can be followed in real-time at the single-nucleotide level. And yet this precision introduces hundreds of degrees-of-freedom as genetic changes accrue in parallel lineages over generations. On short timescales, physiological constraints have been leveraged to provide a coarse-grained view of bacterial gene expression characterized by a small set of phenomenological parameters. Here, we ask whether this same framework, operating at a level between genotype and fitness, informs physiological changes that occur on evolutionary timescales. Using a strain adapted to growth in glucose minimal medium, we find that the proteome is substantially remodeled over 40 000 generations. The most striking change is an apparent increase in enzyme efficiency, particularly in the enzymes of lower-glycolysis. We propose that deletion of metabolic flux-sensing regulation early in the adaptation results in increased enzyme saturation and can account for the observed proteome remodeling.

Adaptive laboratory evolution provides a controlled environment within which genetic change can be tracked with single-nucleotide precision[1,2]. Despite the depth of data, there is no consensus framework for predicting the trajectory of adaptation, nor de-convolving how the genomic changes confer increased fitness. Part of the challenge comes from the large number of mutations that accumulate over the course of the experiment, and the possible interdependence of each of these mutations on all others in the evolutionary history of the organism[3,4]. Yet the recurrence of a common collection of mutations in parallel lineages hints at a set of simplifying principles[5,6].

The longest-running, and most well-studied, adaptation experiment was begun by Lenski in 1988[7], using 12 founding lineages of *Escherichia coli* grown in minimal medium containing a mixture of glucose and citrate. Every 24 h, the cells are diluted 1:100 into fresh media. Here, we focus on the early adaptation of one of these lineages (Ara-1)[2], up to 40,000 generations. The relative fitness (assessed via competition against the ancestral strain over one growth cycle) and the doubling rate of the evolved strains in this lineage both increase monotonically with generation number (Fig. S1). After 20k generations, the genome carries 29 single-nucleotide polymorphisms (SNPs) and 16 deletion-insertion polymorphisms (DIPs). Subsequently, this lineage develops a hypermutator phenotype with dramatically elevated mutation rate so that by 40k generations the genome carries 627 SNPs and 26 DIPs[2]. One striking insertion polymorphism fixed early in this Ara-1 lineage (by 5000 generations[8]) effectively inactivates pyruvate kinase F (*pykF*). Mutations in the *pykF* gene appear in all twelve of the Lenski linages[4]. What makes this mutation remarkable is that pyruvate kinase F catalyzes the final step in glycolysis, converting phosphoenolpyruvate (PEP) to pyruvate. PykF is thought to mediate flux through upper- and lower-glycolysis[9], and its product, pyruvate, is one of the ketoacids that is used to coordinate the carbon catabolic response in *E. coli*[10]. Why mutations in such an important enzyme should appear

[1]Department of Physics, University of California at San Diego, La Jolla, CA, USA. [2]Department of Integrative Structural and Computational Biology, The Skaggs Institute for Chemical Biology, The Scripps Research Institute, La Jolla, CA, USA. [3]Department of Chemical Engineering, University of Waterloo, Waterloo, ON, Canada. [4]Waterloo Centre for Microbial Research and the Department of Applied Mathematics, University of Waterloo, Waterloo, ON, Canada. ✉e-mail: mscott@uwaterloo.ca

so ubiquitously and so early in glucose adaptation experiments remains a puzzle. It has been suggested that the diminished function of PykF observed in the Lenski adaptation experiments is beneficial insofar as it redirects PEP to increase the import rate of glucose via the phosphotransferase system (PTS)[11]. We propose that loss of flux-mediation through glycolysis resulting from the inactivation of the *pykF* gene provides an additional benefit of increasing the efficiency of the enzymes in lower glycolysis by increasing enzyme saturation. The loss of *pykF* produces a higher rate of flux through metabolism without increasing the expression of metabolic proteins, thus avoiding a major physiological constraint on bacterial gene expression[12].

In *E. coli*, one primary physiological constraint is the near-constant total protein concentration[13,14]. As a consequence, if one protein increases in concentration (or, equivalently, in protein mass fraction), then the mass fraction of other proteins must decrease to accommodate the change. In response to translation inhibition, for example, ribosome and ribosome-affiliated proteins increase in mass fraction, with a concomitant decrease in the mass fraction of most other proteins[12,15,16]. Similarly, metabolic limitations (including catabolic limitation via permease titration[10] and anabolic limitation via titration of the key amination enzymes GDH[10] or GOGAT[16]) serve to define other protein groups responding in concert, resulting in a coarse-grained partitioning of the proteome[16,17]. The exponential growth rate can be decomposed as a flux balance among these proteome sectors, effectively treating each sector as a single enzyme with a lumped catalytic constant that quantifies how strongly changes in the sector protein abundance affect the growth rate[16]. Taken together, the constraint on total protein concentration and the coarse partitioning of the proteome define a set of physiological rules that direct the response of the organism to metabolic challenges[10,12,17].

Metabolic enzyme activities are coordinated by adjusting enzyme abundance and by modulating enzyme saturation via substrate concentration[18,19]. For a given enzyme/substrate pair, there is a characteristic concentration of substrate at which half of the enzyme is actively converting substrate to product[20]. In *E. coli*, most enzymes catalyzing irreversible reactions draw upon substrate pools that are close to the half-saturation concentration[20]. There are, however, groups of enzymes in central carbon metabolism that catalyze reversible reactions strongly biased against product formation—for example, the enzymes in lower-glycolysis GapA, Pgk, and GmpA ($K_{eq} = 0.6$, $3.7 \times 10^{-4}$, 0.2, respectively)[21]. In order to move flux in the direction of product formation, these enzymes operate in a regime close to substrate saturation so that changes in flux elicit large changes in substrate concentration. Substrate concentration then serves to amplify changes in flux, and can therefore act as a high-fidelity flux-sensor[22,23]

One of the putative flux-sensing metabolites upstream of GapA is fructose bisphosphate (F1,6BP), which is implicated in the coordination of glycolytic flux by modulating the activity of PykF. We propose that the inactivation of the pykF gene, which removes the actuator from the F1,6BP/PykF flux-sensing mechanism, leads to an increase in intermediate substrate concentrations and, consequently, higher enzyme saturation. As a result of this increase in enzyme efficiency, a smaller concentration of enzyme can be used to carry the flux, thereby freeing up precious space in the proteome.

Here, we use translation limitation and quantitative proteomics to determine how adaptation to growth in glucose over the course of 40k generations remodels the proteome of strains in Lenski's laboratory evolved Ara-1 lineage. We find that adaptation results in an increase in enzyme efficiency, possibly mediated by an increase in substrate saturation. Many of the enzymes exhibiting the largest increase in efficiency lie directly upstream of the *pykF* deletion. Using a simplified mathematical model, we propose that the loss of the flux-sensing mechanism coupling F1,6BP to PykF expression could explain the observed increase in enzyme efficiency. Proteome partitioning

constraints suggest that abrogation of a flux-sensing mechanism provides large fitness gains in a constant nutrient environment, and could provide a general strategy for adaptation.

## Results

### Adaptation of the ribosome-affiliated protein fraction

In the ancestral strain (REL606), and other wildtype strains of *Escherichia coli*, there is a positive linear correlation between the ribosome abundance and the doubling rate when the doubling rate is modulated by the nutrient quality of the medium (e.g., changing carbon source, supplementing with amino acids, etc.)[15,24,25]. The lineage used in this study (Ara-1) cannot metabolize citrate, and so effectively, the cells are adapted to growth in glucose (and glucose metabolic by-products) over the course of 40k generations. We wondered whether this restricted nutrient environment would affect the coupling between the ribosome abundance and doubling rate that was observed in the ancestral strain. It did not; the growth rate dependence of the ribosomal abundance in the ancestral and 40k-adapted strain (10938) are indistinguishable under nutrient-modulated growth rate change (Fig S2A, Supplementary Data 1). Converted to units of ribosome-affiliated protein mass fraction (Fig. S2B), that growth rate dependence is shown in Fig. 1A as a gray line.

The positive linear correlation between ribosome abundance and doubling rate under nutrient-modulated growth (Fig. 1A, gray line) is a consequence of the catalytic role that the ribosome plays in protein synthesis[26]; the intercept $R_0$ corresponds to a fraction of inactive ribosomes awaiting charged tRNA[27] (Fig. 1B, pale green) and the remainder corresponds to an active fraction $\Delta R$ that is proportional to the rate of protein synthesis[15,24] (Fig. 1B, dark green). In Fig. 1B, $\Delta R^*$ corresponds to the active ribosome fraction in the 40k strain. The Lenski adaptation protocol results in a monotone increase in the active ribosome fraction (Fig. 1B, center panel) that is commensurate with the increase in the growth rate of the adapted strain.

There are many ways to modulate the doubling rate of *E. coli*. A particularly useful method is translation inhibition using sublethal concentrations of a ribosome-targeting antibiotic[12]. When grown in sublethal concentrations of antibiotic, the protein synthesis flux is decreased as ribosomes are inactivated[24]. In response, ppGpp signaling increases the synthesis of new ribosomes to compensate for the translation inhibition[28], resulting in a negative linear correlation between the ribosome abundance and the doubling rate (Fig. 1A), irrespective of the chemical details of the antibiotic[15,24,25]. In contrast to nutrient-modulated growth, translation limitation reveals a clear difference between the ancestral strain (Fig. 1A, black circles) and the 40k-adapted strain (Fig. 1A, purple stars): the vertical intercept increases substantially after adaptation, with no obvious change in the slope.

The ribosomal proteins occupy a large fraction of the proteome under most growth conditions[16,17]. Translation-limitation increases the ribosome-affiliated protein mass fraction, and as a consequence of the proteome partitioning constraints, necessarily decreases the protein mass fraction of non-ribosomal proteins (including metabolic enzymes)[12]. We can then use the ribosome abundance at maximum inhibition to infer the abundance of non-ribosomal proteins under nominal (i.e., antibiotic-free) growth conditions (Fig. 1B, orange). This growth-dependent non-ribosomal protein fraction is composed primarily of metabolic enzymes, and so we denote it by $\Delta M$ (and $\Delta M^*$ in the 40k-adapted strain). What is remarkable about the translation-limited response of the adapted strain is that the active metabolic protein fraction increases with an increase in the antibiotic-free growth rate ($\Delta M^* > \Delta M$). This is a clear difference between modulation of the nominal growth rate via nutrient quality as compared to modulation via adaptation: in a given strain, the maximum ribosome abundance under translation limitation is largely growth medium independent[15,24] (see also Fig. S1C), and so the active metabolic fraction decreases with

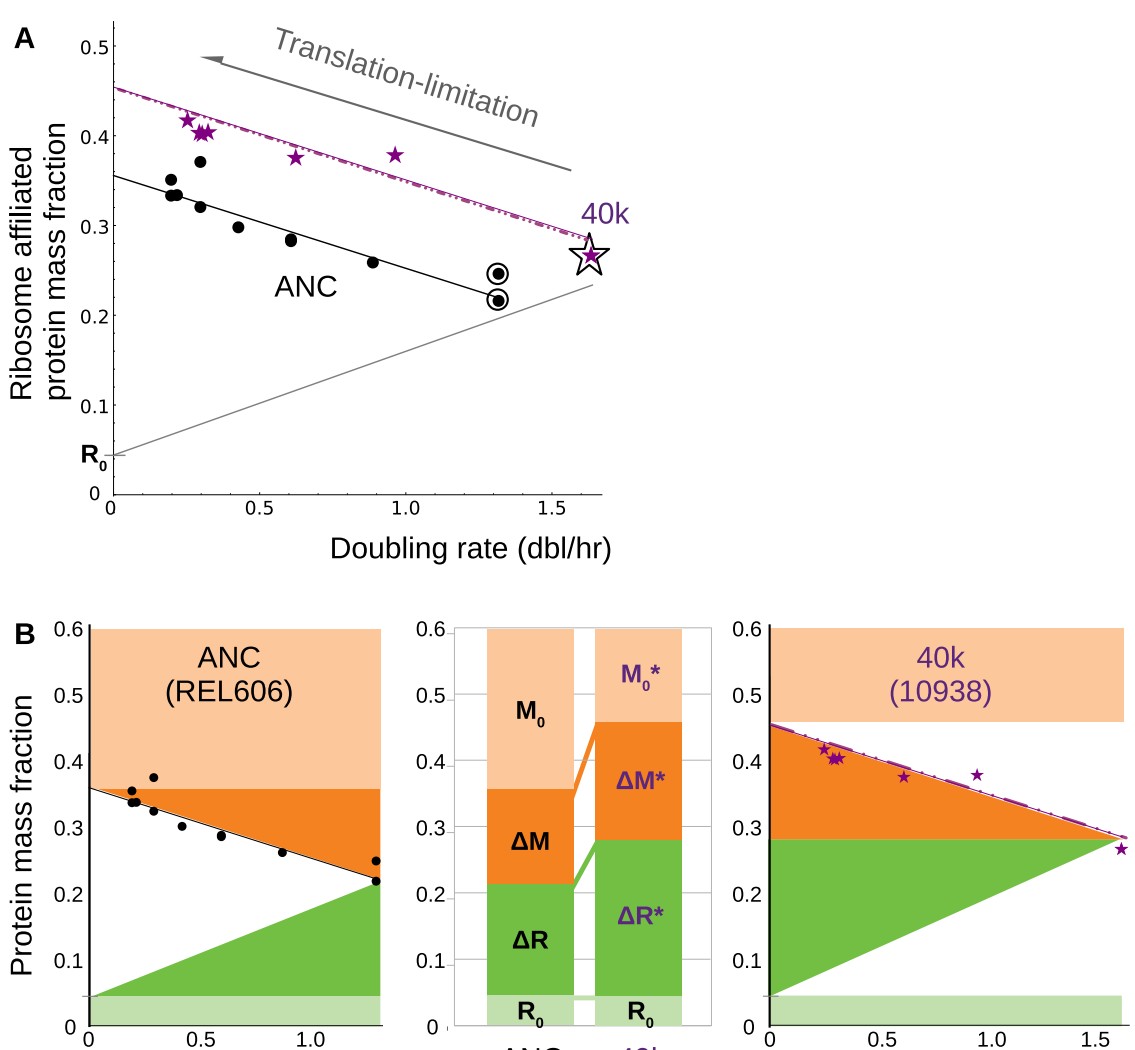

**Fig. 1 | Quantifying the effect of adaptation on ribosome abundance using translation limitation. A** The gray line denotes the change in ribosome abundance under changes in nutrient quality; the ancestral and 40k strains exhibit the same slope and intercept (Fig. S2A, Supplementary Data 1). Using chloramphenicol to inhibit translation, the ribosome-affiliated protein mass fraction increases linearly with decreasing doubling rate in both the ancestral strain (REL606, black) and the 40k-adapted strain (10938, purple). The bacteria are grown in a glucose-minimal medium (outlined symbols correspond to antibiotic-free growth). There is a substantial increase in the intercept of the linear fit for the adapted strain and no appreciable change in slope (Supplementary Note 1). **B** The ribosome abundance serves to partition the proteome into four coarse-grained sectors: active ($\Delta R$, green) and inactive ($R_0$, pale green) ribosomal proteins, and active ($\Delta M$, orange) and inactive ($M_0$, pale orange) non-ribosomal (e.g., metabolic) proteins. Adaptation appears to increase both active sectors ($\Delta R$ and $\Delta M$), with a commensurate decrease in the inactive metabolic protein fraction ($M_0$). Here, and throughout, the star (*) denotes the 40k-adapted strain.

increasing nutrient-modulated growth rate. Below (cf. Fig. 4) we contend that the orange fraction in Fig. 1B corresponds to the protein mass fraction of active (flux-carrying) metabolic enzymes $\Delta M$, and that the remainder of the proteome is occupied by the inactive fraction of metabolic enzymes $M_0$ (Fig. 1B, pale orange).

In the ancestral strain (and other laboratory wildtype strains), the slope of the linear growth rate dependence in the ribosomal protein mass fraction under translation limitation correlates with the nutrient quality of the medium[15]. The increase in the active metabolic protein fraction and the negligible change in the slope of the translation-limited response of the ribosome-affiliated protein fraction in the 40k-adapted strain both suggest that, at a coarse-grained level, adaptation does not result in glucose being perceived as a better nutrient source (in the same way that, for example, glucose is perceived as a better nutrient source than glycerol in the ancestral strain). To determine the origin of the observed increase in the active metabolic fraction, we next looked directly at the translation-limited abundances of the metabolic proteins.

## Increase in the active fraction of the proteome sectors

Previous work established a coarse-grained partitioning of the proteome based upon response to growth limitations[16]. The partitioning consists of six sectors that exhibit increased protein fraction under translation limitation (R), catabolic limitation (C), anabolic limitation (A), both catabolic and anabolic limitation (S), a growth rate independent sector (O), and the growth rate-dependent, but limitation independent, sector (U). In addition to a shared physiological response, proteins in each sector share common metabolic roles (Fig. 2A).

For the ancestral and 40k-adapted strains under translation-limited growth, all sectors of the proteome exhibit a linear growth rate dependence (Fig. 1A and Fig. 2B–F), that serves to partition each sector into active and inactive fractions (for example, denoted by $\Delta A^*$ and $A0^*$ in Fig. 2B). In the adapted strain we observe a change in the magnitude of the active and inactive fractions, though we observe no obvious change in the slope of the linear behavior; a linear-regression constrained to have the same slope between the ancestral and 40k data is

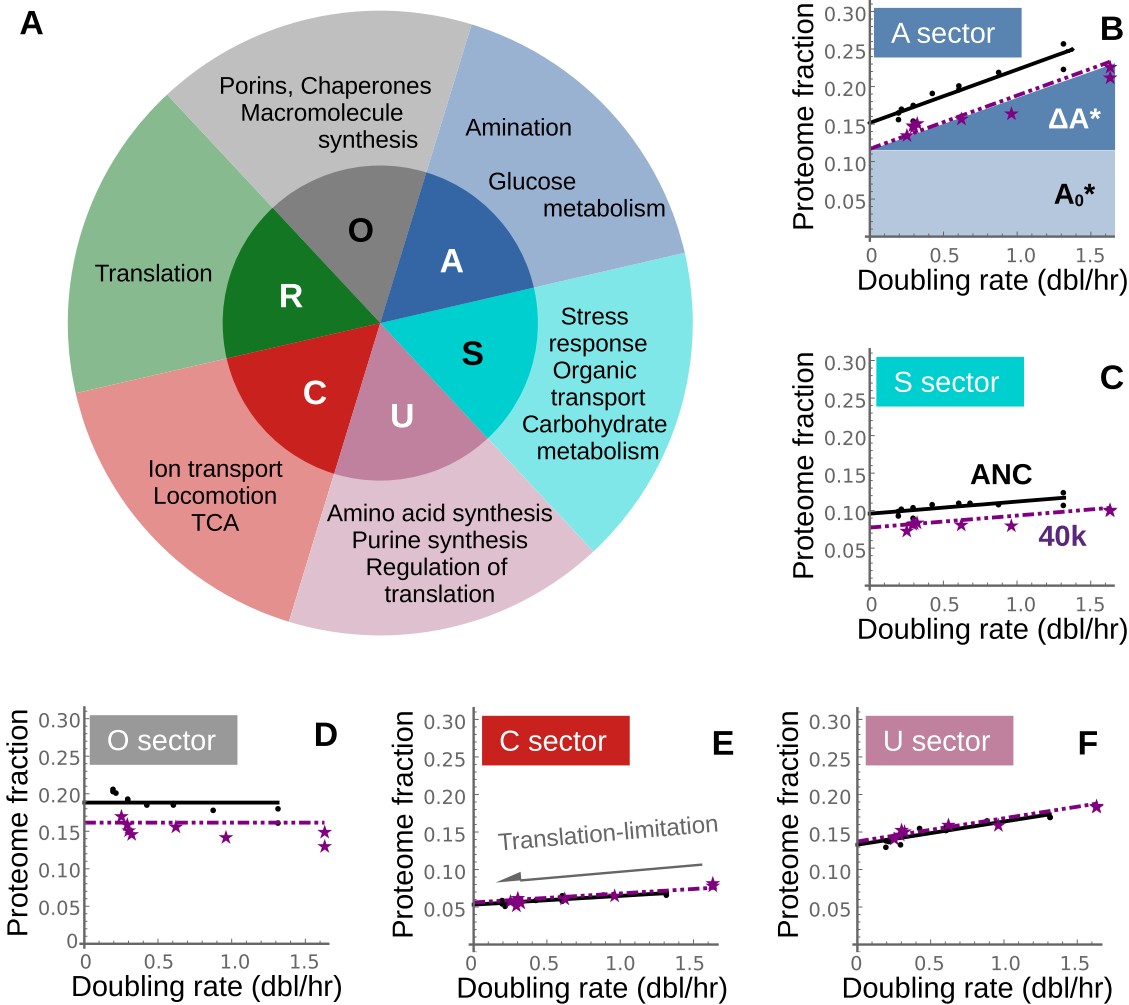

**Fig. 2 | Quantifying the effect of adaptation on proteome partitioning using translation limitation. A** Partitioning of the proteome based upon physiological response to growth inhibition results in six sectors, with shared function among proteins in each sector[4]. Under translation-limited growth, the protein mass fraction of each sector exhibits a linear growth rate dependence, with an intercept corresponding to the inactive protein fraction (e.g., $A_0^*$ in **B**), and the remainder corresponding to an active protein fraction (e.g., $\Delta A^*$ in **B**). The data across all sectors is well-fit by assuming no change in slope between the ancestral (black) and 40k strain (purple) (Supplementary Note 1). **B**, **C**) In the A- and S-sectors, we observe a decrease in the inactive protein fraction (i.e., a decrease in the intercept) after adaptation. **D** The growth rate independent O-sector exhibits decreased expression levels in the adapted strain. **E**, **F** There is no change in the inactive protein fraction for the C and U sectors, although both exhibit a slight increase in the active fraction after adaptation. The black lines are best-fit to the ancestral data; the purple dashed lines are the best-fit across the 40k strain assuming no change in slope (such that the intercepts across all six sectors sum to 1). The sum of active fractions in panels B-F are equal to their respective active fraction ($\Delta M$ or $\Delta M^*$) shown in orange in Fig. 1B (see Fig. S3).

nearly as descriptive as an unconstrained linear-regression model (constrained average $r^2 = 0.82$ as compared to the unconstrained average $r^2 = 0.83$; see Supplementary Note 1).

We observed in the 40k strain that changes in the translation-limited behavior of the non-ribosomal proteome is sector-specific. There is a decrease in the inactive fraction of both the A- and S-sectors after adaptation (Fig. 2B, C). There is a decrease in the abundance of the growth rate-independent O-sector in the adapted strain (Fig. 2D). The C- and U sectors lie upon the same translation-limitation line, with no change in the inactive fraction and a slight increase in the active fraction after adaptation (Fig. 2E, F) (Supplementary Data 1). The combined behavior of the non-ribosomal protein sectors is consistent with the two-sector partitioning shown in Fig. 1B insofar as the active fraction of each sector increases after adaptation (Fig. S3).

**Individual proteins recapitulate sector behavior**

We checked whether the observed variation in sector behavior held for the behavior of individual proteins within each sector. The expressed proteins vary in abundance over several orders of magnitude, so as a proxy for the slope of the translation-limited response, Fig. 3A plots the relative change in the protein abundance at 1.3 dbl/h versus the relative change in abundance at the zero-growth intercept. For all sectors, the data is clustered along the diagonal (Fig. 3A, filled circles denote the average) consistent with sector response (Fig. 3A, crosses), and indicative of little change in slope for the translation-limited response of individual proteins between the ancestral and adapted strains. Those proteins lying in the upper-right quadrant have a linear response shifted upward after adaptation (similar to the R-protein sector data shown in Fig. 1A); whereas proteins lying in the lower-left quadrant have a linear response shifted downward after adaptation (similar to the A- and S-protein sector data shown in Fig. 2B, C) indicating a reduced inactive protein fraction in the 40k strain.

The decrease in the O-sector is primarily due to a regulated decrease in the abundance of the porin OmpF[29]. Figure 3B provides a list of the absolute changes in the inactive protein fraction between the ancestral and 40k strains of individual genes in the S- and A-sectors, as inferred from the intercepts of the linear, fits shown in Fig. 2B, C. The genes are selected to provide a minimal set accounting for at least 50%

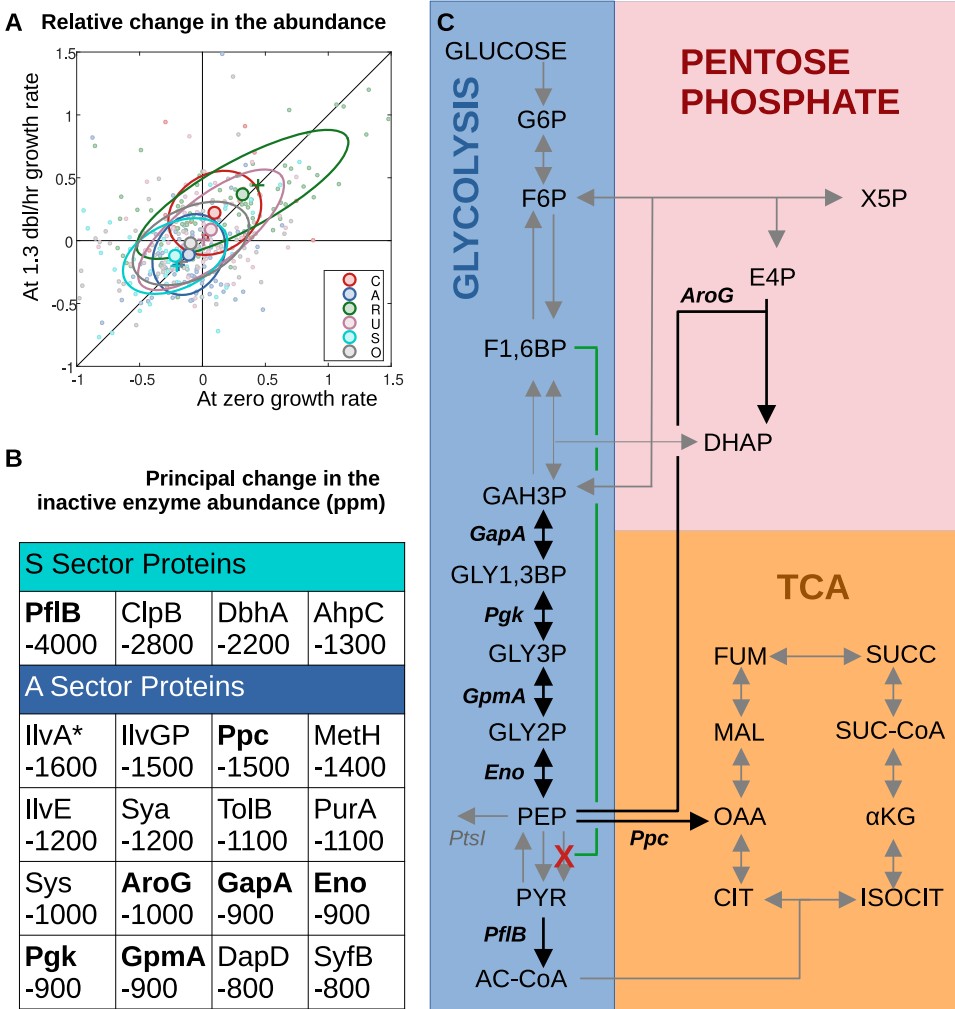

**A** Relative change in the abundance

**B** Principal change in the inactive enzyme abundance (ppm)

| S Sector Proteins | | | |
|---|---|---|---|
| **PflB** -4000 | ClpB -2800 | DbhA -2200 | AhpC -1300 |
| A Sector Proteins | | | |
| IlvA* -1600 | IlvGP -1500 | **Ppc** -1500 | MetH -1400 |
| IlvE -1200 | Sya -1200 | TolB -1100 | PurA -1100 |
| Sys -1000 | **AroG** -1000 | **GapA** -900 | **Eno** -900 |
| **Pgk** -900 | **GpmA** -900 | DapD -800 | SyfB -800 |

**Fig. 3 | Protein expression changes adjacent to the PykF deletion. A** The relative change in the nominal protein fraction (normalized to growth at 1.3 dbl/h) is compared to the relative change in abundance extrapolated to zero-growth. Those points lying on the diagonal exhibit no change in slope upon adaptation. Large circles correspond to the average over each sector, along with one-standard-deviation ellipses. The crosses correspond to the sector fits shown in Fig. 2. Low abundance proteins (<400 ppm) were not included in the figure, and represent less than 15% of the detected protein fraction. **B** A minimal set of genes that explain 50% of the observed change in the inactive protein fraction of the S- and A-sector proteins. The numbers correspond to the absolute change in translation-limited intercept protein mass fraction between the ancestral and 40k strains (expressed as parts-per-million). Plots of the individual proteins appear in Fig. S4. **C** Enzymes involved in carbon metabolism that exhibit the largest change in inactive protein fraction (bold in **B**) are found immediately up- and down-stream of the pyruvate kinase F deletion (PykF, red cross). The second downward arrow adjacent to the PykF deletion corresponds to the reaction catalyzed by PykA, a weak isozyme of PykF that likewise converts PEP to pyruvate. PykF is the target of a flux-sensing mechanism conveyed by small-molecule regulation that uses the concentration of F1,6BP as a proxy for the flux through upper glycolysis[14,15] (green line). The metabolic map was redrawn from the KEGG database[43].

of the observed change in the inactive fraction of each sector. The changes in the S-sector proteins are confined to a shortlist, including the glycolytic enzyme PflB. By contrast, the A-sector proteins comprise small changes distributed among many genes. It is important to note that after 40k generations, almost none of these genes carry mutations in their coding region or their promoters—the single exception is IlvA which carries a single-nucleotide polymorphism at amino acid 124 (Phe→Cys)[2]. This suggests that the observed decreases in the inactive enzyme fractions for these proteins are not of proximal genetic origin, but rather are the result of distal genetic changes conveyed through regulation and biochemistry. We sought a potential molecular mechanism for generating the observed decrease in the inactive enzyme abundance that does not include direct mutation of the enzymes themselves.

More than a third of the S- and A-sector proteins responsible for the principal change in the inactive enzyme abundance are clustered in the metabolic map near the highly-abundant, high-flux-carrying protein pyruvate kinase F (*pykF*; red cross, Fig. 3C) which is deleted early in the adaptation[2,8]. Pyruvate kinase F is the target of a flux-sensing mechanism whereby the concentration of fructose bisphosphate (F1,6BP) correlates with the flux through upper glycolysis and activates PykF in response to an increase in flux[9,30] (Fig. 3C, green line). Between the metabolite F1,6BP and its putative target PykF are reversible reactions, with GapA, Pgk, and GpmA biased to favor the substrate[23] (Fig. 3C, bold). It is this bias that makes the concentration of F1,6BP a high-fidelity reporter of glycolytic flux[23], and we suggest that it is precisely this same strong bias that produces the observed decrease in the inactive enzyme abundance of the upstream enzymes upon *pykF* deletion.

**Mechanism for proteome remodeling after adaptation**
The translation-limited behavior of the proteome sectors can be rationalized by considering the behavior of an irreversible enzyme. For a simple enzymatic reaction, with the substrate in excess of the enzyme (that is, the enzyme-limited regime), the rate of product formation is given by the familiar Michaelis–Menten expression (Fig. 4A). There is a

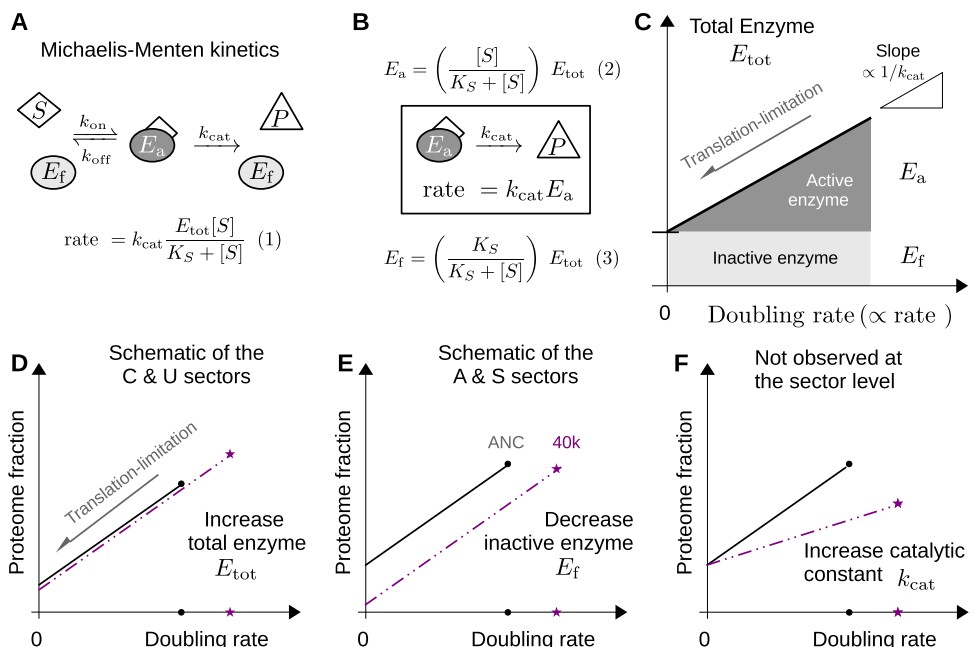

**Fig. 4 | Linearity in the proteome fractions under translation limitation. A** For an irreversible enzymatic reaction, the rate of product formation is proportional to the concentration of enzyme-engaged inactive complex with the substrate, $E_a$. If the substrate concentration $[S]$ far exceeds the total concentration of enzyme $E_{tot} = E_a + E_f$, then the active complex concentration $E_a$ takes the familiar Michaelis–Menten form ref. 34. **B** Distinguishing between the active complex $E_a$ and the inactive, free-enzyme $E_f$ provides a rationalization for the proteomic data. **C** For many enzymatic reactions, rate of product formation is proportional to the doubling rate[20]. In *E. coli*, inhibiting translation attenuates the doubling rate without affecting the catalytic constant $k_{cat}$[3,4]. Consequently, the translation-limited response of an individual enzyme provides an estimate of the active flux-carrying enzyme abundance $E_a$, the inactive free-enzyme abundance $E_f$ (intercept), and the catalytic constant $k_{cat}$ (reciprocal slope). **D** Adaptation in the C and U sectors is consistent with an increase in the total enzyme abundance $E_{tot}$. **E** In contrast, adaptation in the A- and S-sectors is consistent with a decrease in the inactive free-enzyme abundance $E_f$. **F** At the coarse-grained level, we did not observe any change in the slope of the translation-limited response, which would be consistent with an increased catalytic efficiency $k_{cat}$.

dynamic equilibrium between the substrate-bound active complex $E_a$ and the (inactive) free-enzyme $E_f$, but the rate of product formation is strictly proportional to the active complex concentration (Fig. 4B).

Previous work has quantified subtle changes in protein expression in the adapted strains by using 2D gels[31] or ribosome density[32]. Our approach differs in that coupling proteomics with translation limitation allows us to estimate what fraction of the expressed protein is actively flux-carrying. For many reactions, the flux is proportional to the doubling rate[33]: cells growing twice as quickly must convert metabolic intermediates twice as quickly. In *E. coli*, translation limitation provides a method for modulating the doubling rate (and therefore the reaction fluxes) by modulating the protein synthesis rate without changing the base growth medium and without affecting the catalytic constants of the metabolic enzymes[16]. Consequently, as the doubling rate is decreased via translation inhibition, the abundance of individual enzymes decreases in linear proportion by directly attenuating the abundance of active (flux-carrying) enzyme $E_a$ (Fig. 4C). The intercept of the translation-limited response corresponds to the inactive free-enzyme abundance $E_f$, and the slope is proportional to the reciprocal of the catalytic constant $k_{cat}$.

The translation-limited response of a given proteome sector is a weighted average of the enzymes that make up that sector[34]. After 40k generations of adaptation, the doubling rate is increased, necessitating increased flux through each sector; three scenarios are shown schematically in the lower panels of Fig. 4, illustrating how a flux increase could be achieved. For the catabolic C- and U-sector proteins, the translation-limited behavior is consistent with an increase in flux mediated primarily by increasing the total protein abundance $E_{tot}$ at the drug-free growth rate (Fig. 4D, compare the black circle to the purple star). If the inactive free-enzyme abundance (intercept) and the catalytic constant (slope) remain unchanged, an increase in the total enzyme abundance directly increases the active flux-carrying fraction $E_a$.

By contrast, the biosynthetic and glycolytic S- and A-sector proteins exhibit translation-limited behavior that is consistent with a decrease in the inactive free-enzyme abundance $E_f$ (intercept) without an apparent change in the total enzyme abundance at the drug-free growth rate $E_{tot}$ or a change in the catalytic constant (slope) (Fig. 4E). Strong selective pressure directed toward increasing the flux through a single reaction can result in mutations that improve enzyme performance[35,36], which would appear as a reduced slope (increase in $k_{cat}$) when plotting the enzyme levels against the growth rate (Fig. 4F), yet this behavior was not observed in any of the coarse-grained sectors.

For the C/U-sector response, a linear increase in the total enzyme abundance with growth rate can be generated by a regulatory mechanism that forces a positive correlation between the substrate concentration and the total enzyme concentration, as is typical of substrate-driven feed-forward activation[12,22,37]; either at the level of individual genes or via a master regulator[9].

A regulatory mechanism that maintains the linear growth rate dependence in both the ancestral and adapted strains, but with a decrease in the inactive free-enzyme abundance, as exhibited by the A- and S-sector proteins (Fig. 4E), is more difficult to imagine. Based upon the changes in the individual inactive enzyme fractions for enzymes directly upstream of the PykF deletion, we wondered whether the abrogation of the flux-sensing mechanism (Fig. 3C, green line) could explain the observed adaptation in the A- and S-sector proteins. Deletion of PykF could yield increased substrate saturation of upstream enzymes (and thereby decrease the inactive free-enzyme fractions) because these enzymes catalyze thermodynamically unfavorable reactions[23,38]. PykF generates a forward driving force for these reactions that is controlled by upper glycolytic flux via the

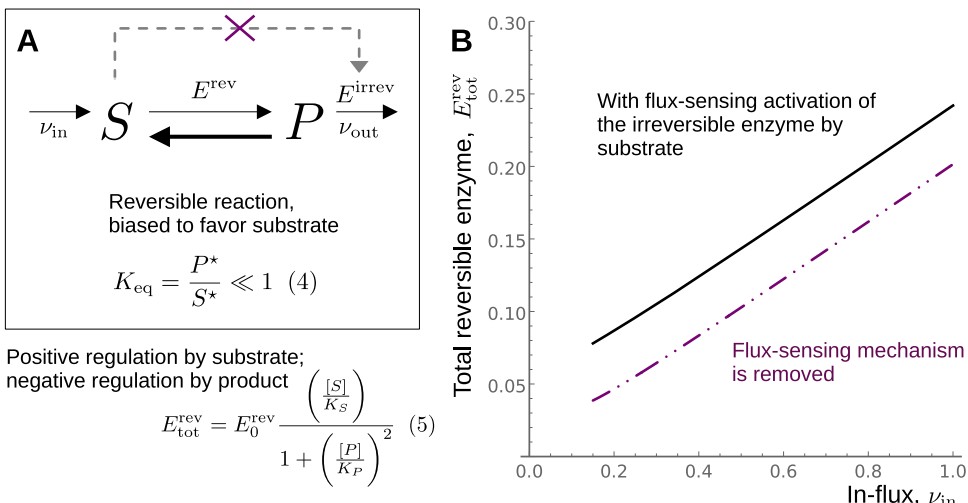

**Fig. 5 | Removal of a flux sensor can reduce the inactive free-enzyme pool. A** A toy model that contains the essential features of the F1,6BP/PykF flux-sensor[14,33]. The intervening reversible reaction (catalyzed by the enzyme $E^{rev}$) is strongly biased in favor of the substrate—that is, in the absence of in-flux $v_{in}$, the equilibrium concentration of substrate $S^*$ far exceeds the equilibrium concentration of product $P^*$. We assume that the total enzyme abundance catalyzing this reversible reaction $E_{tot}^{rev}$ is positively regulated by substrate and negatively regulated by product. The flux-sensing metabolite activates one of the enzymes responsible for the out-flux $v_{out}$ (dashed arrow), and this flux-sensing mechanism is removed in the 40k strain (purple cross). **B** The total enzyme abundance for the reversible reaction $E_{tot}^{rev}$ exhibits a linear increase with the in-flux $v_{in}$ (black). Deletion of the flux-sensor results in the same linear response, but with decreased intercept (purple), consistent with the observed behavior of the A- and S-sector proteins (Fig. 2B, C). Details of the model are included in Supplementary Note 2.

concentration of the metabolite F1,6BP[9]. PykF deletion decouples this driving force-flux relationship, possibly increasing the saturation of the intermediate enzymes. We constructed a toy model to assess whether the deletion of a flux-sensing mechanism could reproduce the decrease in the inactive enzyme fraction (but with negligible change in the linearity or the slope) as exhibited by the A- and S-sector proteins (Fig. 2E, F).

An overview of the model network is shown in Fig. 5A. The target of the flux-sensing metabolite (with concentration $[S]$) is the enzyme $E^{irrev}$ catalyzing the irreversible consuming reaction with flux $v_{out}$ (Supplementary Note 2). The intervening reactions between the source and target of the flux sensor are represented by a single reversible reaction catalyzed by the enzyme $E^{rev}$. The strong correlation between the in-flux $v_{in}$ and the substrate concentration $[S]$ requires a strong bias in favor of the substrate in the reversible reaction, $K_{eq} \ll 1$[23], and is a necessary condition for the substrate to act as a flux-sensor. We posit that the abundance of the enzyme catalyzing the reversible reaction, $E_{tot}^{rev}$, is mediated by a simple regulatory motif consisting of activation by the substrate $[S]$ and feedback inhibition by the product $[P]$.

For sufficiently high in-flux $v_{in}$, the model operates in the enzyme-limited regime, and the Michaelis–Menten forms for the reversible and irreversible kinetics are appropriate[39]. In this regime, the total abundance of the enzyme catalyzing the reversible consumption of substrate, $E_{tot}^{rev}$, is linearly correlated with the in-flux $v_{in}$ (Fig. 5B, black). Furthermore, the removal of the flux-sensitivity in the irreversible consumption reaction decreases the inactive enzyme abundance (intercept) without affecting the slope of the line (Fig. 5B, purple), consistent with the observed behavior in the A- and S-sector proteins after adaptation (Fig. 2B, C). In all, the simple model suggests that abrogation of flux-sensing mechanisms could be generically beneficial during adaptation as a means of partly relieving proteome allocation constraints by increasing enzyme efficiency through substrate saturation.

## Discussion

The concentration of total protein in *E. coli* is nearly growth rate-independent[13,14], and this imposes strong anti-correlations among highly-expressed proteins[12]. In response to translation limitation, for example, the ribosome-affiliated protein fraction increases linearly with decreasing growth rate to occupy 35–45% of the proteome (Fig. 1A), necessarily decreasing the fraction of non-ribosomal proteins. We have taken advantage of this anti-correlation by using translation limitation as a means of globally decreasing active flux-carrying metabolic enzyme abundance without perturbing the nutrient environment.

For strains adapted to growth in glucose over 40k generations[2], we find that the growth rate increases monotonically with generation number. To produce an increase in the growth rate during adaptation, we observe that the active flux-carrying fractions of ribosome-affiliated and metabolic proteins increase (Fig. 1B). The constraint that the protein concentration remains unchanged necessitates that proteome space is made available to accommodate the increase in these active flux-carrying fractions by decreasing the inactive enzyme fraction.

By considering the translation-limited response of hundreds of expressed proteins, we find that part of that decrease in the inactive enzyme fraction is attributed to enzymes directly upstream of the pyruvate kinase F (PykF) deletion that occurs early in the adaptation of this lineage. PykF is activated by the upstream flux-sensing metabolite F1,6BP[9]. Strong backward bias in the intervening reversible reactions is a thermodynamic prerequisite for the flux-sensing ability of F1,6BP[23], and so deletion of PykF could decrease the inactive enzyme fraction of the intervening enzymes by increasing substrate saturation.

We created a toy model to assess the plausibility of this scenario (Fig. 5A), and we find that, indeed, the translation-limited response observed in the glycolytic enzymes can be reproduced (Fig. 5B) provided that the transcription of the reversible enzyme is activated by substrate and repressed by product. For the F1,6BP/PykF system, that regulation could be mediated by the transcriptional regulators Cra and cAMP-Crp. The Cra protein represses the expression of GapA, Pgk, GpmA, and Eno[40], and is itself negatively correlated with F1,6BP concentration[9]; consequently, F1,6BP effectively acts as an activator for the expression of the reversible enzymes upstream of PykF and F1,6BP levels are increased upon PykF deletion[41]. The global regulator cAMP-Crp activates the expression of GapA and Pgk[40], and although Crp is not directly responsive to PEP, PykF deletion leads to a high conversion flux of PEP to oxaloacetate (OAA)[41], which is one of several

ketoacids that inhibit cAMP synthesis via adenylate cyclase[10]. Consequently, a potential source of transcriptional regulation by product feedback inhibition would be OAA inhibiting cAMP-Crp, thereby reducing the transcription of GapA and Pgk. If antagonistic substrate activation/product repression motifs are identified in other putative flux-sensing systems, then abrogation of flux-sensors could provide a generic adaptation response that creates space in the proteome by increasing enzyme saturation and flux without increasing enzyme concentration.

For growth in glucose, the PykF deletion can be rationalized by the proteome remodeling that occurs in its absence, although it is less clear whether this deletion has any direct benefit when the strains are grown on other substrates[42]. The diminished function (or occasionally deletion) of pyruvate kinase F (pykF) is commonly observed in the adaptation of E. coli to growth in glucose minimal media[2,4,6], and in the directed evolution of isobutyrate[43] and L-serine[44] producing strains. Consistent with our observed decrease in the non-flux-carrying intercepts of the A- and S-sector proteins, deletion of pykF does increase glycolytic substrates (notably glucose-6-phosphate and fructose-6-phosphate), as well as reducing cellular pyruvate levels[41]. Because pyruvate is one of the primary reporters of carbon glycolytic flux in E. coli[10,30], we expect the loss of pykF in the Lenski strains to result in impaired carbon catabolite repression.

The translation-limited proteomic response is a tool with application outside of the coarse-grained physiological characterization we have done. Applied to individual enzymes, the interpretation outlined in the upper panels of Fig. 4 provides a framework for estimating in vivo enzymatic activity. Taking the ratio of the flux to the total enzyme concentration, the enzyme activity is $k_{cat}/(1 + E_f/E_a)$, where all three parameters are fully determined by the translation-limited response. At present, the in vivo activity of individual metabolic enzymes is estimated using isotope tracing for the flux and quantitative proteomics for protein abundance. If reaction flux and enzyme activity inferred from the translation-limited behavior of individual proteins is validated by metabolic measurements, then quantitative proteomics could be used to provide a convenient, all-in-one complementary methodology to isotope tracing.

Framing adaptation dynamics in terms of a coarse-grained physiological model can be applied more generally to other microorganisms, including engineered minimal genome strains[45]. In the Lenski-adapted strains, the translation-limited intercepts exhibit a relative change with a generation number that is about 100x more rapid than corresponding changes in the doubling rate. As a result, significant remodeling of the proteome can be observed on a timescale of several weeks of adaptation, rather than several decades, allowing for comparatively-convenient exploration of a variety of laboratory evolution scenarios.

Our focus has been on the physiological consequences of adaptation manifest in exponential growth. The Lenski strains spend about seven hours a day in that state, the remaining time is spent transitioning in and out of stationary phase. A complete picture of physiological adaptation to this growth regimen would necessarily include these other growth states. Nevertheless, the observed changes in proteome partitioning can be largely understood in terms of adjusting metabolic flux through changes in protein expression following deletion events. Further, these proteomic changes could not be deduced from the available genome sequences during adaptation, highlighting the importance of the expressed proteome as an energetically costly carrier of metabolic flux.

## Methods
### Growth of bacterial culture
**Bacterial strains.** All strains used in this study come from the Ara-1 lineage of the Lenski long-term evolution experiment[2]. The ancestral strain is REL606; the adapted strains are pure-strains isolated after several thousand generations of growth: 2k (REL1164A), 10k

(REL4536A), and 40k (REL10938). These strains have been fully sequenced[2].

**Growth medium.** All growth media used in this study were MOPS-buffered based upon Neidhardt[46], and obtained commercially (Teknova, M2101). Carbon sources used were glycerol (0.2% v/v) and glucose (0.2% w/v). For $^{15}$N-labeled media, $^{15}$NH$_4$Cl was used in place of $^{14}$NH$_4$Cl (Teknova, M2120 with 20 mM $^{15}$NH$_4$Cl (Sigma)). For nutrient-modulated growth (Fig. S2A), the minimal medium was enriched using 0.2% (w/v) casamino acids (Fisher), or nucleotides and amino acids (Teknova, M2103, and M2104) as indicated in Supplemental Data 1 (denoted by CAA [casamino acids] and RDM [rich defined medium], respectively). Translation-limited growth was achieved by adding chloramphenicol to the growth medium at sublethal concentrations (0–12 μM) as indicated in Supplemental Data 1. Susceptibility to chloramphenicol is inversely related to the antibiotic-free growth rate[47], making the 40k strain more susceptible than the ancestral. The maximum concentration of chloramphenicol was chosen so that the growth rate was between 0.1–0.2 doublings/hour.

**Growth measurements.** All batch culture growth was performed in a 37 °C water bath shaker shaking at 250 rpm. The culture volume was at most 5 ml in 25 mm × 150 mm test tubes. Each growth experiment was carried out in three steps: a seed culture in LB broth, pre-culture, and experimental culture in an identical minimal medium. For seed culture, one colony from a fresh LB agar plate was inoculated into liquid LB and cultured at 37 °C with shaking. After 4–5 hrs, cells were centrifuged (17,000 × g) and washed once with an appropriate minimal medium. Cells were then diluted into the minimal medium and cultured in 37 °C water bath shaker overnight (pre-culture). The overnight pre-culture was allowed to grow for at least three doublings. Cells from the overnight pre-culture were then diluted to OD$_{600}$ = 0.005–0.025 in identical pre-warmed minimal medium and cultured at 37 °C in a water bath shaker (experimental culture). 150 μl cell culture was collected in a Hellma microvolume 10 mm quartz cuvette (Hellma, Mullheim) for OD$_{600}$ measurement using a Thermo BioMate3S Spectrophotometer around every half doubling of growth. 5–7 OD$_{600}$ data points within the range of ~0.05 and ~0.5 were used for calculating the growth rate.

### Protein and RNA quantification
**Total protein quantitation.** The Biuret method was used for total protein quantitation[48]. Briefly, 1.8 ml of exponentially growing cell culture at around OD$_{600}$ = 0.5 was collected by centrifugation (17,000 × g). The cell pellet was washed with phosphate-buffered saline, re-suspended in 0.2 ml phosphate-buffered saline, then stored at 4 °C overnight. The cell pellet was brought up to room temperature. 0.1 ml of 3 M NaOH was added to the cell pellet and samples were incubated at 100 °C heat block for 5 min to hydrolyze proteins. Samples were then cooled to room temperature. The Biuret reactions are carried out by adding 0.1 ml 1.6% CuSO$_4$ to samples with thorough mixing, and the color is allowed to develop over 5 mins. Samples were centrifuged for 3 min at 17,000 × g to clear the solution, and the absorbance at 555 nm was measured by spectrophotometer. The same Biuret reaction was applied to a series of BSA (bovine serum albumin) standards to generate a standard curve.

**Total RNA quantitation.** The RNA quantitation method is based on the method used by Benthin et al.[49], with modifications. Briefly, 1.5 ml of cell culture at around OD$_{600}$ = 0.5 during the exponential phase was collected by centrifugation (17,000 × g). The cell pellet was immediately washed twice with 0.6 ml cold 0.1 M HClO$_4$, and stored at 4 °C overnight. The next day, the pellets were brought to room temperature, then digested with 0.3 ml 0.3 M KOH for 60 min at 37 °C with constant shaking. The cell extracts were neutralized with 0.1 ml 3 M HClO$_4$ and centrifuged at 17,000 × g for 5 min. The supernatant was

collected, and the precipitate was washed twice with 0.55 ml 0.5 M $HClO_4$. A final volume of 1.5 ml of supernatant was then centrifuged for 10 mins at $17,000 \times g$ to clear the solution, and the supernatant absorbance at 260 nm was measured by spectrophotometer. The RNA concentration ($\mu g/ml/OD_{600}$) was calculated by $OD_{260} \times 31/OD_{600}$, where we have used the conversion factor of 31 between the $OD_{260}$ and RNA concentration, based upon the molar extinction coefficient of $10.5\ \text{mmole}^{-1}\ \text{cm}^{-1}$ and the average molecular weight of an *E. coli* RNA nucleotide residue of 324.

## $^{15}$N-labeled proteomic mass spectrometry

**Sample preparation.** 1.8 ml of cell culture at $OD_{600} = 0.4\sim0.5$ during the exponential phase of the experimental culture was collected centrifugation ($17,000 \times g$). The cell pellet was re-suspended in 0.2 ml water and fast-frozen on dry ice. Samples were assigned random IDs to preclude any unintended bias in the analysis.

Aliquots of the $^{15}$N reference cell sample (or labeled cell sample) were mixed with each of the $^{14}$N cell samples (or non-labeled cell samples). Each aliquot of the $^{15}$N and $^{14}$N samples contained 100 μg of protein. For each growth condition and strain, the $^{15}$N reference cell sample was made from a 1:1 mixture of protein extracted from the ancestral strain (REL606) grown in glucose minimal medium and from the 40k strain (REL10938) grown in glucose minimal medium with the maximum concentration of chloramphenicol (12 μM). The mixed reference is used to avoid bias in the composition of proteins in the reference cell sample by a particular growth condition and background mutation.

The sample preparation prior to mass spectrometry was as described in detail in Hui et al.[16]: Proteins were precipitated by adding 100% (w/v) trichloroacetic acid (TCA) to 25% final concentration. Samples were placed on ice for a minimum of 1 h. The protein precipitates were pelleted by centrifugation ($16,000 \times g$) for 10 min at 4 °C. The supernatant was removed, and the pellets were washed with cold acetone. The pellets were dried in a Speed-Vac concentrator.

The dry pellets were dissolved in 80 μl of 100 mM $NH_4HCO_3$ with 5% acetonitrile (ACN). 8 μl of 50 mM dithiothreitol (DTT) was added to reduce the disulfide bonds before the samples were incubated at 65 °C for 10 min. Cysteine residues were modified by the addition of 8 μl of 100 mM iodoacetamide (IAA) followed by incubation at 30 °C for 30 min in the dark. The proteolytic digestion was carried out by the addition of 8 μl of 0.1 μg/μl trypsin (Sigma-Aldrich, St. Louis, MO) with incubation overnight at 37 °C.

The peptide solutions were cleaned by using the PepClean C18 spin columns (Pierce, Rockford, IL). After drying in a Speed-Vac concentrator, the peptides were dissolved into 10 μl sample buffer (5% ACN and 0.1% formic acid).

**Mass spectrometry.** The peptide samples were analyzed on an AB SCIEX TripleTOF 5600 system (AB SCIEX, Framingham, MA) coupled to an Eksigent NanoLC Ultra system (Eksigent, Dublin, CA). The samples (2 μl) were injected using an autosampler. Samples were loaded onto a Nano cHiPLC Trap column 200 lm × 0.5 mm ChromXP C18-CL 3 lm 120 Å (Eksigent) at a flow rate of 2 μl/min for 10 min. The peptides were separated on a Nano cHiPLC column 75 μm × 15 cm ChromXP C18-CL 3 lm 120 Å (Eksigent) using a 120-min linear gradient of 5–35% ACN in 0.1% formic acid at a flow rate of 300 nl/min. MS1 settings: mass range of m/z 400–1250 and accumulation time 0.5 s. MS2 settings: mass range of m/z 100–1800, accumulation time 0.05 s, high sensitivity mode, charge state 2–5, selecting anything over 100 cps, maximal number of candidate/cycle 50, and excluding former targets for 12 s after each occurrence.

**Protein identification.** The raw mass spectrometry data files generated by the AB SCIEX TripleTOF 5600 system were converted to Mascot generic format (mgf) files, which were submitted to the Mascot database searching engine (Matrix Sciences, London, UK) against the *E. coli* SwissProt database to identify proteins. The following parameters were used in the Mascot searches: maximum of two missed trypsin cleavage, fixed carbamidomethyl modification, variable oxidation modification, peptide tolerance ±0.1 Da, MS/MS tolerance ±0.1 Da, and 1+, 2+, and 3+ peptide charge. All peptides with scores less than the identity threshold ($p = 0.05$) were discarded.

**Relative protein quantitation.** The raw mass spectrometry data files were converted to the.mzML and.mgf formats using conversion tools provided by AB Sciex. The.mgf files were used to identify sequencing events against the Mascot database. Spectra for peptides from the Mascot search were quantified using least-squares Fourier transform convolution implemented in-house[50]. Data were extracted for each peak using a retention time and m/z window enclosing the envelope for both the light and heavy peaks. The data are summed over the retention time, and the light and heavy peaks amplitudes are obtained from a fit to the entire isotope distribution, yielding the relative intensity of the light and heavy species. The ratio of the non-labeled to labeled peaks was obtained for each peptide in each sample.

The relative protein quantitation data for each protein in each sample mixture was then obtained as a ratio by taking the median of the ratios of its peptides. No ratio (*i.e.*, no data) was obtained if there was only one peptide for the protein. The uncertainty for each ratio was defined as the two quartiles associated with the median. To filter out data with poor quality, the ratio was removed for the protein in that sample if at least one of its quartiles lay outside of the 50% range of its median. Ratios were removed for a protein in all the sample mixtures if at least one of the ratios has one of its quartiles lying outside of the 100% range of the median. Because the ratios are all defined relative to the same reference sample, they represent the relative change of the expression of the protein across all the non-labeled cell samples and are referred to as relative expression data.

**Absolute protein quantitation.** The spectral counting data used for absolute protein quantitation were extracted from Mascot search results. For our $^{15}$N and $^{14}$N mixture samples, only the $^{14}$N spectra were counted. The absolute abundance of a protein was calculated by dividing the total number of spectra of all peptides for that protein by the total number of $^{14}$N spectra in the sample. These peptide counts are recorded in Supplemental Data 1.

### Reporting summary
Further information on research design is available in the Nature Portfolio Reporting Summary linked to this article.

## Data availability
Raw mass spectral data is deposited to massIVE, with the accession code MSV000087313 or available through the Proteomexchange (http://www.proteomexchange.org/) via the accession code PXD025666. All other data are available in the Supplementary Information.

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

## Acknowledgements
All strains were graciously provided by Richard Lenski. We are grateful to Fernanda Pinheiro, Meriem el Karoui, and Terence Hwa for their comments on the manuscript. This work is supported by the NIH through grants R35GM136412 (J.R.W. and V.P.) and R35GM152133 (M.M.) and by NSERC through grant 2024-04038 (M.S.).

## Author contributions
Conceptualization, investigation, and writing—original draft: M.S. (growth, RNA, and protein), M.S. and M.M. (data analysis), V.P and J.R.W. (quantitative proteomics), C.E., M.S. (mathematical modeling). Writing—review and editing: M.S., J.R.W., C.E., and M.M. All authors reviewed the results and approved the final version of the manuscript.

## Competing interests
The authors declare no competing interests.
