## [Peer Review File · Nature Communications]

Proteome partitioning constraints on long-term laboratory evolutionREVIEWER COMMENTS

Reviewer #1 (Remarks to the Author):

I understand that the search for coherence in physiological data is important, even essential, if we are to make progress in our understanding of bacterial function (including evolution).

But, as always, the question is whether we are there yet, not that it is not interesting to analyze the data in new ways, but can we not already do it a little better?

In this context, putting the analysis of protein distribution at the heart of the analysis of bacterial physiology is relatively recent, and is due to recent progress in the fine quantification of proteomic concentrations (about ten years ago).

The question now is to analyze the different proteins and their behavior. Obviously, defining large sectors is a possibility (as in the article by Hui et al. 2015, here, the authors start from the idea that disruption of certain protein sectors disrupts others (this is already a feature of the article by Hui et al. 2015)).

In parallel with such a global analysis, contributions in the literature as proposed a new mathematical method to predict a significant part of the bacterial cell components, such as protein concentrations and ribosome concentrations and specific features as the growth rate, in different physiological conditions.

This prediction has also been made in a more recent article on *E. coli*.

In fact, using complete (but simplified) models of the cell (such as those that exist for *E. coli*, for example), it is now possible to understand effectively and rigorously (within the assumptions of the models developed) that the disruption/restriction of a bacterial function can, through this restriction, lead to changes in the functioning of the cell. Two types of questions can then be examined. The first is based on the adaptation of the regulatory network to the perturbation, making it possible to anticipate and understand what the bacteria are likely to do in the face of the perturbation. Although this first question is interesting (and important from the point of view of understanding the regulatory network), it is difficult and complex because the regulatory network, even in *coli*, is not completely known and mastered. Then there's the important question of evolution and the appearance of mutations, which can also radically change the way things work and are not really predictable.....

Even if we consider a longer time frame, the question of the weight of the changes is central, and we can imagine that a mutation with a strong impact on physiology is the most likely... we still have to define what that means...

The second option is a little trickier, because it is based on the idea that evolution and natural selection will tend to modify the bacteria so that 1) they respect the constraints but 2) they are still 'efficient' under those new constraints. But clearly the question is more difficult because it presupposes that we know how to 'predict'/model the bacteria finely enough to be able to predict what is/are the most favorable configurations under this constraint...

Although the study carried out by the authors is interesting in itself, this document does not seem to me to be up to date in terms of knowledge and practice. At the very least, and if this is what the authors think, they should show that existing theories are unable to deal with the issue in detail. Indeed, it is always possible not to know about other approaches, but to ignore them seems to me to be a real problem, because if these methods don't make it possible to deal with this question, this should be shown and explained.

Reviewer #2 (Remarks to the Author):

The presented manuscript by Mori et al. reports on making use of quantitative proteomics and proteome partitioning to determine how adaptation of laboratory evolved *E. coli* strains (isolates from the Lenski evolution experiment) is reflected in proteome partitioning. The authors find the increase in growth rate over adaptation to be differently accounted for by different proteome sectors to enable increased fluxes.

While this work of Mori et al. is very timely and the study tackles an important question (how can we gain a better understanding of the trajectories of adaptation in evolution experiments), there are in my opinion some critical conceptual points that need to be clarified and revised before publication.

Major Comments:

1. The authors claim that making use of the described method of R-limitation and proteome partitioning can yield valuable insights to better understand the trajectories of bacterial adaptation that are currently untapped by genomic and physiological approaches. While I believe that this holds true and I value the described findings, I am unsure if the manuscript in its current stage outlines the strengths and convincing findings of the described experiments sufficiently. While the authors partially discuss the strengths and limitations of this approach, I think they could make a stronger case for the impact of this work and work out the knowledge gap and the novelty of their findings more. Explicitly: in the discussion work out more clearly in what way your work advances the previous results on the Lenski isolates.
2. Along these lines, generally the manuscript lacks structure and the rational and reasoning is unfortunately hard to follow. In several instances, paragraphs are not well connected, data presentation and interpretation in the text lacks clearness and transitions between the subsections of the results are not well motivated and approaches not sufficiently described. To make the work accessible to a broader audience I would advise the authors to reconsider the manuscript's narrative, clearly motivate and describe the chosen methodological approaches and to derive a stronger argumentation throughout the manuscript (why is this important, what is the explicit question, how to answer it). While there are several instances to improve the structure of the manuscript overall, I would explicitly urge the authors to revise the introduction to highlight what is the state of the art, what are the new ideas, strength, hypotheses of the presented work, as well as the discussion. Here, a possibility would be to move the first section of the discussion including Fig. 4 into the results part, as in my opinion this is central for the argumentation of the presented approach and for the reader to fully apply the proposed concept to interpret the data.
3. A central hypothesis of the described research is that the addition of chloramphenicol results in ribosome inhibition (and hence translation inhibition) and therefore, when comparing proteome partitioning of different strains, changes in the remaining proteome fraction can be fully attributed to changes that occurred due to adaptation. Does the literature and/or potentially performed controls provide sufficient evidence that this holds true and that the application of chloramphenicol really is sufficiently specific and acts comparable on all strains? In other words, how can we be sure, that potentially resulting stress responses of differently evolved strains do not contribute to "higher order" changes in the proteome that are not directly linear across adapted strains? Please comment on this and potentially integrate this in the discussion of the manuscript.

4. All samples were obtained in exponential growth phase of the different strains; how exactly was this determined, and how was it normalized across strains and conditions (as e.g. doubling time decrease with increasing chloramphenicol conc. does not look identical for all strains)?

Unfortunately, the methods description on this (and generally) is quite sparse and in the SI data only doubling times are available. While I would ask the authors to clearly describe their approach, it might also be helpful to make growth data accessible. This would help to provide insights into how overall growth changes with the application of chloramphenicol (in terms of yield, lag, etc.).

5. As shown in Fig. 2 and described in the text L151ff, none of the proteome sectors change in slope – only intercepts change (increase for R-sector, decrease for O, S, A-sector). Further, the authors state in the Figure legend of Fig. 2 that there was “little change observed for C and U sectors”, while I would argue that there is no change observed at all. Would it be possible to include other, more quantitative, approaches to determine the observed changes? Same for Fig. 3A. Please comment on this and potentially include more quantitative data analysis here.

Minor Comments:

- Please add more information to the methods section (as described above) but also concerning statistical methods, data analysis, etc.
- L39/40: is there experimental proof of this? If so, please cite.
- L42ff and L52/53 this is directly related – the information in these paragraphs is quite scattered; as described above, please make more clear and accessible
- L73ff: this is really important background info. Make the setup of the work more clear and also include more information

Reviewer #3 (Remarks to the Author):

In this work, authors Mori, Patsalo, Williamson, and Scott describe their research contribution to a long-standing question: how does the physiology of bacteria change over the course of adaptive evolution? While there is a plethora of genetic observations exist, little is known about the physiologically relevant changes. Given the constraints, a microorganism can increase its fitness by either boosting the abundance of a limiting enzyme or by increasing its catalytic ability. The former appears simplest to implement, it can only occur at the expense of some other protein, while the latter--increasing enzyme's catalytic activity--requires alteration of the enzyme structure, with no cost of additional expression. Using evolved strains from long-term evolution experiment from Lenski laboratory, the authors identify that the increased growth rate of evolved strains arises from the expression of metabolic enzymes undergoing sector-specific adaptation. This manuscript suggests a method--exploiting R-limitation in conjunction with proteomics--that could be applied to look for adaptation patterns and connect the proteomic composition to genetic changes.

Manuscript is thus timely, relevant, and would represent a welcome addition to knowledge.

Data and methods are well documented and presented, and one can follow the reasoning. A researcher, skilled in the art, could reproduce the experimental approaches as the cornerstone of the experimental work is clearly outlined.

While the manuscript is of high quality, I would recommend the authors to consider the following points:

Larger points:

- The data in the manuscript show clear trends in the proteome allocation of the adapted strain. I think the article would gain further credibility if the authors would look closer into genetic changes that might underlie the observed changes. Authors indeed mention some of the changes (deletion

of pykF, and gltBD); I would suggest the authors check the genetic underpinnings of the genes whose expression changed most dramatically. Is there any common trend in promoter regions? Is the coding region altered (check for possible mutations of the known catalytic sites? If so, is there a pattern there (rarer codons, weaker start codons, ...)?

- Large changes in the expression patterns can be achieved by many small changes, as well as alteration of activity/expression of major regulators. I would further recommend authors closely inspect possible changes in the regulation of expression of major regulators and regulatory components (such as CRP [and cAMP synthesis/degradation machinery], relA, spoT, ..., as well as rpoS). Changes in there could suggest attractive hypotheses about the evolution of large-scale changes in the proteomic constitution of bacterial cells. The absence of such patterns is equally interesting, as it suggests that smaller, directed climbs along the fitness landscape occur in place of the larger ones.

Minor

- Fig 1: standard deviation requires at least three observations to be evaluated.

- Personally, I do not find the choice of mathematical notation to be the most informative.

Specifically, $\Delta R/M$ currently used to denote the dynamical part of the proteome would be better suited to describe the dynamical range (as one of the authors used it before in Greulich, Scott, et al [2015, Mol System Biol]).

- While I appreciate that the number of protrusions in the symbols denotes the increasing number of generations that passed, the gray symbols in Fig 1 are hard to read in the absence of plot marker outlining. Further in Fig 2, bigger symbols would serve presentation better. In Fig 1. The changed carbon source is already denoted by the hue of the symbols--I find the use of dashed lines adds more confusion than clarity to the plot.

The manuscript is written in clear prose that is easy to follow. The text contains a reasonable amount of referrals to other resources and clearly outlines the aim, approaches, and findings. I appreciate the clarity. The manuscript is adequately referenced, and I did not find any unsubstantiated claims.

While my expertise is predominately in bacterial physiology and quantitative models describing it, I do not have first-hand experience with analysis of the proteomics data. Based on the previous research record of the authors, I am inclined to believe that the current analyses are of equal competency demonstrated before.

We appreciate the insightful and constructive comments of the reviewers. We have substantially reworked the text in response to their suggestions. Comments are addressed line-by-line below (our responses are in green text).

Reviewer #1 (Remarks to the Author):

I understand that the search for coherence in physiological data is important, even essential, if we are to make progress in our understanding of bacterial function (including evolution). But, as always, the question is whether we are there yet, not that it is not interesting to analyze the data in new ways, but can we not already do it a little better?

In this context, putting the analysis of protein distribution at the heart of the analysis of bacterial physiology is relatively recent, and is due to recent progress in the fine quantification of proteomic concentrations (about ten years ago).

The question now is to analyze the different proteins and their behavior. Obviously, defining large sectors is a possibility (as in the article by Hui et al. 2015, here, the authors start from the idea that disruption of certain protein sectors disrupts others (this is already a feature of the article by Hui et al. 2015)).

In parallel with such a global analysis, contributions in the literature as proposed a new mathematical method to predict a significant part of the bacterial cell components, such as protein concentrations and ribosome concentrations and specific features as the growth rate, in different physiological conditions. This prediction has also been made in a more recent article on E. coli.

In fact, using complete (but simplified) models of the cell (such as those that exist for E. coli, for example), it is now possible to understand effectively and rigorously (within the assumptions of the models developed) that the disruption/restriction of a bacterial function can, through this restriction, lead to changes in the functioning of the cell. Two types of questions can then be examined. The first is based on the adaptation of the regulatory network to the perturbation, making it possible to anticipate and understand what the bacteria are likely to do in the face of the perturbation. Although this first question is interesting (and important from the point of view of understanding the regulatory network), it is difficult and complex because the regulatory network, even in coli, is not completely known and mastered. Then there's the important question of evolution and the appearance of mutations, which can also radically change the way things work and are not really predictable.....

Even if we consider a longer time frame, the question of the weight of the changes is central, and we can imagine that a mutation with a strong impact on physiology is the most likely... we still have to define what that means...

The second option is a little trickier, because it is based on the idea that evolution and natural selection will tend to modify the bacteria so that 1) they respect the constraints but 2) they are still 'efficient' under those new constraints. But clearly the question is more difficult because it presupposes that we know how to 'predict'/model the bacteria finely enough to be able to predict what is/are the most favorable configurations under this constraint...

Although the study carried out by the authors is interesting in itself, this document does not seem to me to be up to date in terms of knowledge and practice. At the very least, and if this is what the authors think, they should show that existing theories are unable to deal with the issue in detail. Indeed, it is always possible not to know about other approaches, but to ignore them seems to me to be a real

problem, because if these methods don't make it possible to deal with this question, this should be shown and explained.

We have added more context to the introduction. The narrative is now focused on the puzzling loss of an important glycolytic enzyme (pyruvate kinase F) that occurs early in the adaptation, and ubiquitously in glucose adaptation experiments. There is a qualitative explanation for this deletion based upon rerouting of PEP through the PTS to increase glucose uptake rate (Woods et al. Proc. Natl. Acad. Sci. **103**: 9107 (2006)), although we know of no evidence establishing glucose uptake as a growth-limiting rate in the ancestral strain. Our explanation is built upon a synthesis of recent work (Dourado et al. PLOS Biol. **19**: e3001416 (2021); and Euler et al. Biophys. J. **121**: 237 (2022)), applied to rationalizing adaptation trajectories. This context appears in L45-56 (highlighted in the main text).

Reviewer #2 (Remarks to the Author):

The presented manuscript by Mori et al. reports on making use of quantitative proteomics and proteome partitioning to determine how adaptation of laboratory evolved E. coli strains (isolates from the Lenski evolution experiment) is reflected in proteome partitioning. The authors find the increase in growth rate over adaptation to be differently accounted for by different proteome sectors to enable increased fluxes.

While this work of Mori et al. is very timely and the study tackles an important question (how can we gain a better understanding of the trajectories of adaptation in evolution experiments), there are in my opinion some critical conceptual points that need to be clarified and revised before publication.

Major Comments:

1. The authors claim that making use of the described method of R-limitation and proteome partitioning can yield valuable insights to better understand the trajectories of bacterial adaptation that are currently untapped by genomic and physiological approaches. While I believe that this holds true and I value the described findings, I am unsure if the manuscript in its current stage outlines the strengths and convincing findings of the described experiments sufficiently. While the authors partially discuss the strengths and limitations of this approach, I think they could make a stronger case for the impact of this work and work out the knowledge gap and the novelty of their findings more. Explicitly: in the discussion work out more clearly in what way your work advances the previous results on the Lenski isolates.

We appreciate your thorough and constructive comments. We have expanded the introduction, and focused the narrative on the proteomics data (supporting RNA/protein ratio data is moved to the supplement). We have included a new figure (Fig. 5) and a toy model in order to clarify the novelty of the findings.

2. Along these lines, generally the manuscript lacks structure and the rational and reasoning is unfortunately hard to follow. In several instances, paragraphs are not well connected, data presentation and interpretation in the text lacks clearness and transitions between the subsections of the results are not well motivated and approaches not sufficiently described. To make the work accessible to a broader audience I would advise the authors to reconsider the manuscript's narrative, clearly motivate and describe the chosen methodological approaches and to derive a stronger argumentation throughout the manuscript (why is this important, what is the explicit question, how to answer it). While there are several instances to improve the structure of the manuscript overall, I would explicitly urge the authors

to revise the introduction to highlight what is the state of the art, what are the new ideas, strength, hypotheses of the presented work, as well as the discussion. Here, a possibility would be to move the first section of the discussion including Fig. 4 into the results part, as in my opinion this is central for the argumentation of the presented approach and for the reader to fully apply the proposed concept to interpret the data.

The reviewer's suggestions are excellent – we have reformulated the narrative along the line suggested.

3. A central hypothesis of the described research is that the addition of chloramphenicol results in ribosome inhibition (and hence translation inhibition) and therefore, when comparing proteome partitioning of different strains, changes in the remaining proteome fraction can be fully attributed to changes that occurred due to adaptation. Does the literature and/or potentially performed controls provide sufficient evidence that this holds true and that the application of chloramphenicol really is sufficiently specific and acts comparable on all strains? In other words, how can we be sure, that potentially resulting stress responses of differently evolved strains do not contribute to “higher order” changes in the proteome that are not directly linear across adapted strains? Please comment on this and potentially integrate this in the discussion of the manuscript.

We have included a more detailed discussion of this point (L126-134). Ribosome-targeting antibiotic (e.g. chloramphenicol, tetracycline, kanamycin, streptomycin), mutants in peptide elongation rate, and mutants in translation-initiation-factors all produce the same increase in ribosome abundance with decreasing growth rate (Scott *et al. Science* **330**: 1099 (2010) has a survey of this historical data as Fig. S2, and a demonstrated-equivalence between chloramphenicol and slow-translating mutants in Fig. 1B), which suggests that it is the inhibition of translation and not the chemical details of the antibiotic that underlies the increase in the ribosomal protein fraction. These details are included in L126-134 (highlighted in the text).

4. All samples were obtained in exponential growth phase of the different strains; how exactly was this determined, and how was it normalized across strains and conditions (as e.g. doubling time decrease with increasing chloramphenicol conc. does not look identical for all strains)? Unfortunately, the methods description on this (and generally) is quite sparse and in the SI data only doubling times are available. While I would ask the authors to clearly describe their approach, it might also be helpful to make growth data accessible. This would help to provide insights into how overall growth changes with the application of chloramphenicol (in terms of yield, lag, etc.).

We have added an expanded methods section to the supplement. Briefly, our cultures are grown in test tubes in 3 mL of growth medium, with aeration by shaking. We grow a seed culture from plate in LB to inoculate an overnight culture in the experimental medium (which would contain chloramphenicol for the translation-inhibition experiments). The next day, the overnight culture is diluted into fresh medium and allowed to double 5-10 times before the optical density is recorded. We verify that the strains are growing exponentially by ensuring that the increase in optical density (scattering at 600 nm) increases linearly with time on a log-scale.

The decrease in doubling rate upon increase in chloramphenicol concentration is not the same in all strains – the half-inhibition concentration exhibits an inverse correlation with the drug free growth rate (see Greulich *et al. Molecular Systems Biology* **11**: 796 (2015); PMID 26146675) making the better-adapted strains generically more susceptible to chloramphenicol. The maximum chloramphenicol concentration used for each strain was chosen so that the growth rate was reduced to 0.1-0.2 dbl/hr.

Because our cultures are not grown in a plate reader, we have data only in the exponential phase. There has been some interesting work done on the growth-yield trade-off in these adapted strains (for example, Novak et al. *American Naturalist* 168: 242 (2006); PMID 16874633), but this falls outside our main focus.

5. As shown in Fig. 2 and described in the text L151ff, none of the proteome sectors change in slope – only intercepts change (increase for R-sector, decrease for O, S, A-sector). Further, the authors state in the Figure legend of Fig. 2 that there was “little change observed for C and U sectors”, while I would argue that there is no change observed at all. Would it be possible to include other, more quantitative, approaches to determine the observed changes? Same for Fig. 3A. Please comment on this and potentially include more quantitative data analysis here.

We have included more details of the fit in the supplement, along with a comparison to an unconstrained fit to all data.

Minor Comments:

- Please add more information to the methods section (as described above) but also concerning statistical methods, data analysis, etc.

These have been added to the supplemental text.

- L39/40: is there experimental proof of this? If so, please cite.

Now L58, references are (Oldewurtel *et al. Proc. Natl. Acad. Sci.* **118**: e2021416118 (2021); Balakrishnan *et al. Science* **378**: eabk2066 (2022)).

- L42ff and L52/53 this is directly related – the information in these paragraphs is quite scattered; as described above, please make more clear and accessible

These have been consolidated and moved to the introduction.

- L73ff: this is really important background info. Make the setup of the work more clear and also include more information

We have moved this discussion into the introduction with expanded context.

Reviewer #3 (Remarks to the Author):

In this work, authors Mori, Patsalo, Williamson, and Scott describe their research contribution to a long-standing question: how does the physiology of bacteria change over the course of adaptive evolution? While there is a plethora of genetic observations exist, little is known about the physiologically relevant changes. Given the constraints, a microorganism can increase its fitness by either boosting the abundance of a limiting enzyme or by increasing its catalytic ability. The former appears simplest to implement, it can only occur at the expense of some other protein, while the latter--increasing enzyme's catalytic activity--requires alteration of the enzyme structure, with no cost of additional expression. Using evolved strains from long-term evolution experiment from Lenski laboratory, the authors identify that the increased growth rate of evolved strains arises from the expression of metabolic enzymes undergoing sector-specific adaptation. This manuscript suggests a

method--exploiting R-limitation in conjunction with proteomics--that could be applied to look for adaptation patterns and connect the proteomic composition to genetic changes.

Manuscript is thus timely, relevant, and would represent a welcome addition to knowledge.

Data and methods are well documented and presented, and one can follow the reasoning. A researcher, skilled in the art, could reproduce the experimental approaches as the cornerstone of the experimental work is clearly outlined.

While the manuscript is of high quality, I would recommend the authors to consider the following points:

Larger points:

- The data in the manuscript show clear trends in the proteome allocation of the adapted strain. I think the article would gain further credibility if the authors would look closer into genetic changes that might underlie the observed changes. Authors indeed mention some of the changes (deletion of *pykF*, and *gltBD*); I would suggest the authors check the genetic underpinnings of the genes whose expression changed most dramatically. Is there any common trend in promoter regions? Is the coding region altered (check for possible mutations of the known catalytic sites? If so, is there a pattern there (rarer codons, weaker start codons, ...)?

This discussion is now included in L208-214 (highlighted in the main text). There is remarkably-little change in the DNA surrounding those genes with most dramatic changes.

It is important to note that after 40k generations, almost none of these genes carry mutations in their coding region or their promoters – the single exception is *IlvA* which carries a single-nucleotide polymorphism at amino acid 124 (Phe → Cys). This suggests that the observed decreases in the inactive enzyme fractions for these proteins are not of proximal genetic origin, but rather are the result of distal genetic changes conveyed through regulation. We sought a potential molecular mechanism for generating the observed decrease in the inactive enzyme abundance that does not include direct mutation of the enzymes themselves.

- Large changes in the expression patterns can be achieved by many small changes, as well as alteration of activity/expression of major regulators. I would further recommend authors closely inspect possible changes in the regulation of expression of major regulators and regulatory components (such as CRP [and cAMP synthesis/degradation machinery], *relA*, *spoT*, ..., as well as *rpoS*). Changes in there could suggest attractive hypotheses about the evolution of large-scale changes in the proteomic constitution of bacterial cells. The absence of such patterns is equally interesting, as it suggests that smaller, directed climbs along the fitness landscape occur in place of the larger ones.

We have focused the narrative of the manuscript on the deletion of pyruvate kinase F that occurs early in the adaptation, and on the changes of highly-expressed protein. The reviewer's suggestion is excellent, and tracking the changes of major regulators and regulatory components will be valuable. Those experiments fall outside the scope of the present manuscript, but we are following that line of inquiry in a future manuscript that specifically considers changes to the carbon catabolite repression system in these adapted strains.

Minor

- Fig 1: standard deviation requires at least three observations to be evaluated.

We have moved Fig 1 to the supplement, and have used the mean-squared-error as an estimate of dispersion. The number of replicates for each experiment is now included in the data table.

- Personally, I do not find the choice of mathematical notation to be the most informative. Specifically, $\Delta R/M$ currently used to denote the dynamical part of the proteome would be better suited to describe the dynamical range (as one of the authors used it before in Greulich, Scott, et al [2015, Mol System Biol]).

The delta-notation is inherited from our previous work on proteomics (e.g. You *et al. Nature* **500**: 301 (2013); Hui *et al. Mol. Syst. Biol.* **11**: 784 (2015)). The connection to Greulich et al. is that the dynamical part of all sectors in the proteome (as defined in this work) sum to give the dynamic range of the ribosome (as defined in Greulich) – this is illustrated in Fig. S3. We have added a main-text figure panel, Fig. 1B, that we hope clarifies the meaning of the delta-notation.

- While I appreciate that the number of protrusions in the symbols denotes the increasing number of generations that passed, the gray symbols in Fig 1 are hard to read in the absence of plot marker outlining. Further in Fig 2, bigger symbols would serve presentation better. In Fig 1. The changed carbon source is already denoted by the hue of the symbols--I find the use of dashed lines adds more confusion than clarity to the plot.

Fixed.

The manuscript is written in clear prose that is easy to follow. The text contains a reasonable amount of referrals to other resources and clearly outlines the aim, approaches, and findings. I appreciate the clarity. The manuscript is adequately referenced, and I did not find any unsubstantiated claims.

While my expertise is predominately in bacterial physiology and quantitative models describing it, I do not have first-hand experience with analysis of the proteomics data. Based on the previous research record of the authors, I am inclined to believe that the current analyses are of equal competency demonstrated before.

We appreciate your careful reading of the manuscript, and your valuable suggestions for improvement.

REVIEWERS' COMMENTS

Reviewer #2 (Remarks to the Author):

I would like to thank the authors for their constructive and thorough replies to my comments. While it was from times hard to follow the implemented changes in the manuscript, as I could not find a full track-change version of the manuscript, the revision has significantly improved the manuscript and the authors have satisfactorily addressed my major concerns. Especially, streamlining of the manuscript and clarifying the study's motivation and experimental setup greatly improved the paper.

Reviewer #3 (Remarks to the Author):

Authors' revised version of the manuscript is improved, both in clarity and precision. I consider that authors have adequately addressed my comments. I have also read the comments and responses to other reviewers' suggestions, and I would be of the opinion that authors have considered them appropriately. I thus recommend the publication of the article.